# 3D Printing Applications in Orthopaedic Surgery: Clinical Experience and Opportunities

Andrea Fidanza [1], Tullio Perinetti [2], Giandomenico Logroscino [1,†] and Michela Saracco [3,*,†]

1    Mininvasive Orthopaedic Surgery, Department of Life, Health & Environmental Sciences, University of L'Aquila, 67100 L'Aquila, Italy; andrea.fidanza@graduate.univaq.it (A.F.); g.logroscino@gmail.com (G.L.)
2    Department of Civil, Construction-Architectural and Environmental Engineering, University of L'Aquila, 67100 L'Aquila, Italy; tullioperinetti@gmail.com
3    Department of Orthopaedics, Fondazione Policlinico Universitario A. Gemelli IRCCS, Roma—Università Cattolica del Sacro Cuore, 00168 Rome, Italy
*    Correspondence: michelasaracco@gmail.com; Tel.: +39-33-8867-9796
†    These authors contributed equally to this work.

**Abstract:** Background: Three-dimensional (3D) printing is a technology capable of creating solid objects based on the reproduction of computerised images. This technology offers revolutionary impacts on surgical practice, especially in prosthetic and traumatological surgery. Methods: 20 patients with proximal humeral fractures were divided into two groups, one of which involved the processing of a 3D model. The model made it possible to plan the positioning and dimensions of the implants. The results were then compared with those obtained according to the usual procedures. We also reported the irreparable case of a custom revision implants acetabular bone loss treated with a 3D-printed, custom-made implant. Results: In the processed 3D proximal humeral fracture series, in the face of time and costs expenses, surgical and X-ray times were shorter than in the control group. On the other hand, there were no differences in terms of blood loss. The patient who underwent acetabular re-prosthetic surgery in a 3B Paprosky bone loss was managed and solved with a 3D-printed, custom-made implant and reported excellent outcomes at a 1 year follow-up. Conclusion: Three-dimensional printing made it possible to create better pre-operative planning in traumatology in order to optimise surgical procedures and outcomes. It also made it possible to deal with large losses of bone stock in prosthetic revision surgery, even when reconstruction may have appeared impossible with traditional implants.

**Keywords:** 3D technology; proximal humeral fracture; pre-operative planning; hip revision; bone-loss; custom-made acetabular cup

## 1. Introduction

Three-dimensional (3D) printing is a technology capable of creating solid objects based on the reproduction of computerised images. In the medical field, in line with the greater advancements and availability of commercial 3D printers, the 3D reconstruction of anatomical models guarantees an accurate view and a clearer approach to orthopaedic deformities, offering revolutionary impacts on surgical practice, especially in prosthetic and trauma surgery [1,2].

Large bone stock loosening and complex fractures are often unmanageable without accurate preoperative planning and with traditional standard implants. In trauma surgery, a 3D-printed anatomical model can provide a direct and interactive visualisation of the fracture characteristics, ensuring preoperative planning and practical training [3,4]. Moreover, 3D-printed models seem to be the best way to get agreement with the experience of the observers [5]. Typically, these models are built layer-by-layer with the print hothead deposing material on the segmented area by working with a special plastic resin, such as acrylonitrile butadiene styrene (ABS) material or polylactic acid (PLA) [5,6]. While ABS is

printed with extruder temperatures of around 240 °C and requires a heated print bed, PLA is more manageable, is fully recyclable as it is a bioplastic derived from corn or sugar cane, and it does not emit toxic fumes during printing.

The 3D plastic model's aim is to improve the diagnosis of fractures and to help the surgeon think ahead about the surgical approach and hardware needed. Three-dimensional printing is currently under development around the world for orthopaedic disorders, congenital paediatric orthopaedic problems, or tumours [7–9].

In reconstructive prosthetic surgery, 3D custom-made printed implants make it possible to solve the most complicated cases. In the past, these patients were destined to endure disabling treatments without the possibility of reconstruction. The application of 3D printing technology allows the creation of customised systems, with satisfactory functional results and a good safety profile [10].

Herein, we report two of our experiences with 3D-printing technology: the use of 3D -printed technology for planning in the treatment of complex proximal humeral fractures (Section 1) and a case report of a massive bone loss in hip prosthetic surgery treated with a 3D custom-made implant (Section 2).

## 2. 3D-Printed Reproduction of Fractures

### 2.1. Materials and Methods

Twenty consecutive patients treated surgically for complex proximal humeral fractures (Neer classification 3–4) were included in this randomised case-control study [11]. All the patients underwent X-rays and CT scans in the emergency room and were randomly divided into two groups. Patients in Group A (10 cases) were investigated more deeply by a previous, well-described protocol of study [12]: anatomical full-touch/real-size plastic 3D models were made from the CT images to study joint implication, localization, and quality of bone fragments. Patients enrolled in Group B (10 controls) were treated with a standard method without further pre-operative investigations. In fact, once the senior surgeon and the resident fully understood the fracture and predicted how severe the involvement of the vascular component was, they simulated an on-the-table the surgical procedure, planning the correct sequence to reduce fragments, and selecting and applying the most appropriate plate size and its positioning, length, and orientation of the screws.

All the surgical procedures were conducted in the beach chair position, guaranteeing the same surgical team for each procedure. A deltopectoral approach was chosen as it was considered to be the best approach to allow appropriate fracture reduction and fixation of proximal humeral fractures [13].

The Student's t-test has been used to investigate the differences in the means of operative time (from incision to suture of the skin), the number of intraoperative X-rays, and blood loss between Group A and Group B. The values are reported as the means of the results detected by the outcomes studied ± standard deviation. Values were considered significant if $p = < 0.05$, with CI = 95% (confidence interval).

3D Printing of the Real Size Anatomical Model

In Group A, CT scan images were exported in the Digital Imaging and Communication in Medicine (DICOM) format, and these files were imported into InVesalius software (Centre for Information Technology Renato Archer version 3.1,—Ministério da Ciência, Tecnologia e Inovações (MCTI)—Brasil). CT multi-planar sections, spaced at a distance 0.5 mm from each other, were performed to maintain dimensional and detail accuracy in the subsequent stages of processing. In the second step, specific pixels from each 2D image of the CT multiplanar sections were manually selected. These "multiplanar selections" were related only to the bone to be printed. The sum of all the selected pixels in each 2D image have been dealt with in InVesalius as a 3D point cloud. In the third phase, a "3D volume rendering" was carried out displaying only the selected points relative to the bone and a 3D model with oriented surfaces was created from the same 3D point cloud mentioned above. Both the "3D volume rendering" part and the "3D model with oriented

surfaces creation" part were done using InVesalius, following the software's step-by-step workflow In the fourth phase, the 3D model was exported as an .stl file and imported in a second dedicated software (Meshmixer, Autodesk Inc., San Rafael, CA, USA) to check the geometry, to join and/or close open surfaces, and to export the 3D model once again in .stl format.

In the fifth phase, the previously produced .stl file was prepared for 3D printing using the 3D Cura slicing software (Ultimaker Cura, version 4.5.0, Ultimaker B.V., Utrecht, The Netherlands, 2011–2020).

In the sixth phase, the information necessary for the 3D-printing process was exported in .gcode format, with specifical settings required for a "Creality Ender 3 Pro" 3D printer which uses the fused deposition modelling (FDM) additive manufacturing technique for subsequent layers. The thickness of the printing layers was set at 0.3 mm to obtain a good level of detail without lengthening the printing time. The temperature setting of the extruder was 215 °C, and for the printing surface, 60 °C, which followed the indications provided by the polylactic acid (PLA) plastic filament manufacturer, a viable alternative to petroleum-based materials (Spectrum Group Ltd.—Pęcice, Polska). The wall thickness and the infill percentage were respectively set to 0.6 mm and to 100% to ensure a reasonable structural stability for the following insertion of orthopaedic hardware into the model. In the seventh phase, the .gcode file was loaded on the 3D printer and the printing of the white PLA model was started. The printing, layer-by-layer, involved not only the bony parts, but also their temporary support structures (Figure 1A). In the eighth phase, once the construct was detached from the printing surface, the temporary support structures were removed with the aid of pliers, wire cutters, and rigid levers (Figure 1B).

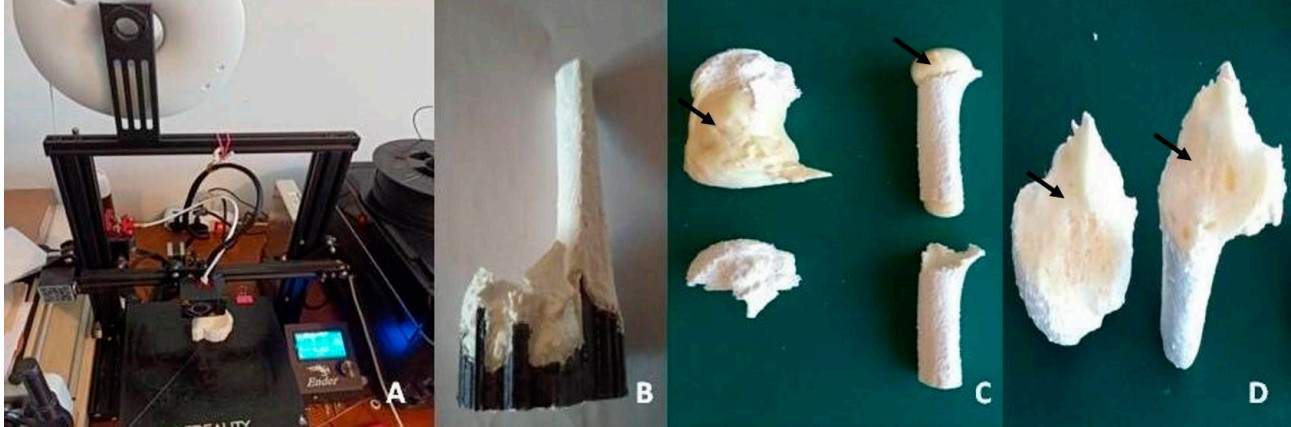

**Figure 1.** The 3D printer (**A**) creates the real-size model with support structures to be removed (in this case, coloured in black to highlight them); (**B**) the polyurethane foam is then applied (black arrows); and (**C**) once dried, the excess parts are removed, simulating cancellous bone and marrow (black arrows) (**D**).

For each of the cases treated, two "full-touch" anatomical models were created: a "fracture model" and a "reduction model". The "fracture model" is a model in which the bone fragments are in the same position as in the CT scan. The "reduction model" is a model in which the bone fragments are printed separately, without any specific regard to their original position in the CT scan.

For the fracture model, in almost all cases treated, it was necessary to introduce "artificial bridges" during the 3D modelling in order to keep the bone fragments in the right position as displayed in the spatial dislocation of the fracture. The 3D models were filled with polyurethane foam in order to roughly reproduce the visual and mechanical characteristics of the cancellous bone (Figure 1C,D).

The anatomical full-touch/full-size PLA models, both as "fracture model" and "reduction model", had undergone a steam sterilisation process at 121 °C for 45 min in order to introduce them into the surgical field [14].

### 2.2. Results

All operations (Group A and Group B) were performed by the same surgical team, consisting of the senior surgeon and two residents.

Planning the surgical procedure for Group A patients was more complex. The average duration of the files processing time from CT scan to 3D printing was 7 h, with a minimum of 4 h in the morphologically less complex case and a maximum of 11 h in a multifragmentary fracture case. The printing time was between 4 and 6 h for each single full-touch anatomical model in full size. Unpretentious costs were recorded for the production of these anatomical full-touch real-size models: less than 5 euros per model as regards the material (PLA) and less than 1000 euros as regards the hardware (commercial laptop and printer). The working hours of processing by a qualified operator were the more expensive cost. The software used is available free of charge for non-commercial purposes. The sterilisation of the 3D models did not involve any additional cost, nor was there a reduction of sterilised instruments to perform the interventions for Group A compared to Group B.

In Group A, the postoperative reduction of the fracture, the size and placement of the plate, the choice of holes to be used, and the direction and length of the screws were superimposed on those carefully planned in the preoperative period, which suggests satisfactory results of the "in Lab" surgical procedure simulation (reduction and fixation of the fracture) on the 3D model (Figure 2). The mean duration of surgery for Group A ($71.28 \pm 10.03$ min) was significantly lower ($p = 0.0194$, CI = 2.319 to 23.181) than in Group B ($84.03 \pm 12.08$ min.), saving about 15% of the surgical time. No statistically significant difference was found in blood loss ($275 \pm 6.25$ dL vs. $282 \pm 10$ dL; $p = 0.0768$, CI = $-0.835$ to 14.835), while a significantly higher number of intraoperative X-rays were recorded in Group B ($11.45 \pm 1.8$ vs. $8.76 \pm 2.1$; $p = 0.0065$, CI = 0.852 to 4.528).

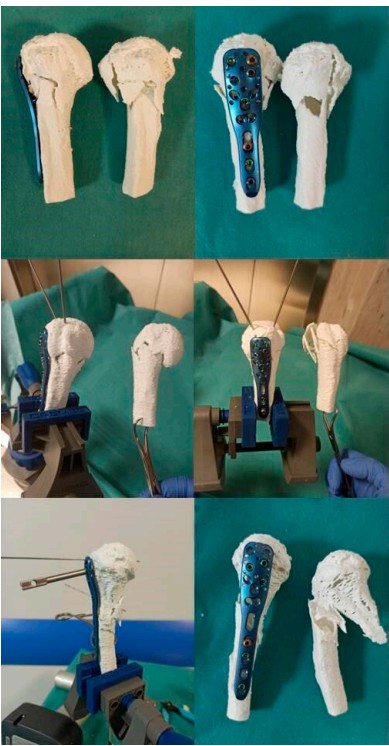

**Figure 2.** Before surgery and after the removal of the artificial bridges (that kept the fragments displaced), a simulation of the reduction and stabilisation of the fracture was performed, allowing the surgeon to plan the best position of the fragments and to choose the best-sized surgical hardware.

### 3. A Dramatic Pelvic Bone Loss

A 44 year-old woman affected by bilateral osteoarthritis secondary to developmental dysplasia of the hip (DDH) underwent a left total hip arthroplasty (THA) in 2006. In this case, a 52 mm cup (Fitmore Shell—Zimmer, Warsaw, IN, USA) with an ultra-high-molecular-weight polyethylene (UHMWPE) liner (Durasul—Zimmer, Warsaw, IN, USA) associated with a conic stem (Conus—Zimmer, Warsaw, IN, USA) and a 28 mm/−4 mm CoCr ball head were implanted.

In 2015, the diagnosis of periprosthetic joint infection (PJI) was made due to the presence of a secerning fistula. Between 2015 and 2018, she underwent six consecutive surgical joint debridements and numerous cycles of antibiotic therapy.

The patient performed clinically well up to November 2019, when she referred with tight pain and a new secerning fistula. Blood examinations were obtained: erythrocyte sedimentation rate (ESR: 36 mm/h; reference values: 0–15 mm/h), C-reactive protein (CRP: 11.6 mg/dL; reference values: <1 mg/dL), and D-dimer (0.8 μg/mL; reference values: <0.4 μg/mL) were elevated. Radio-labelled autologous white blood cells (WBC) scintigraphy was performed, which was positive for PJI.

In January 2020, the first of the "Two-Stage" revision surgeries was performed. This involved prosthesis explant, debridement, and placement of a gentamicin antibiotic spacer (Spacer G, Tecres, Sommacampagna, Italy). The cup was explanted with the aid of a dedicated Zimmer instrument (Explant), while the stem was explanted by a single modified Wagner femoral osteotomy.

The intraoperative bone loss assessment established the loss of the anterior wall of the acetabulum, the fundus, and part of the roof (3B Paprosky classification) [15] (Figure 3).

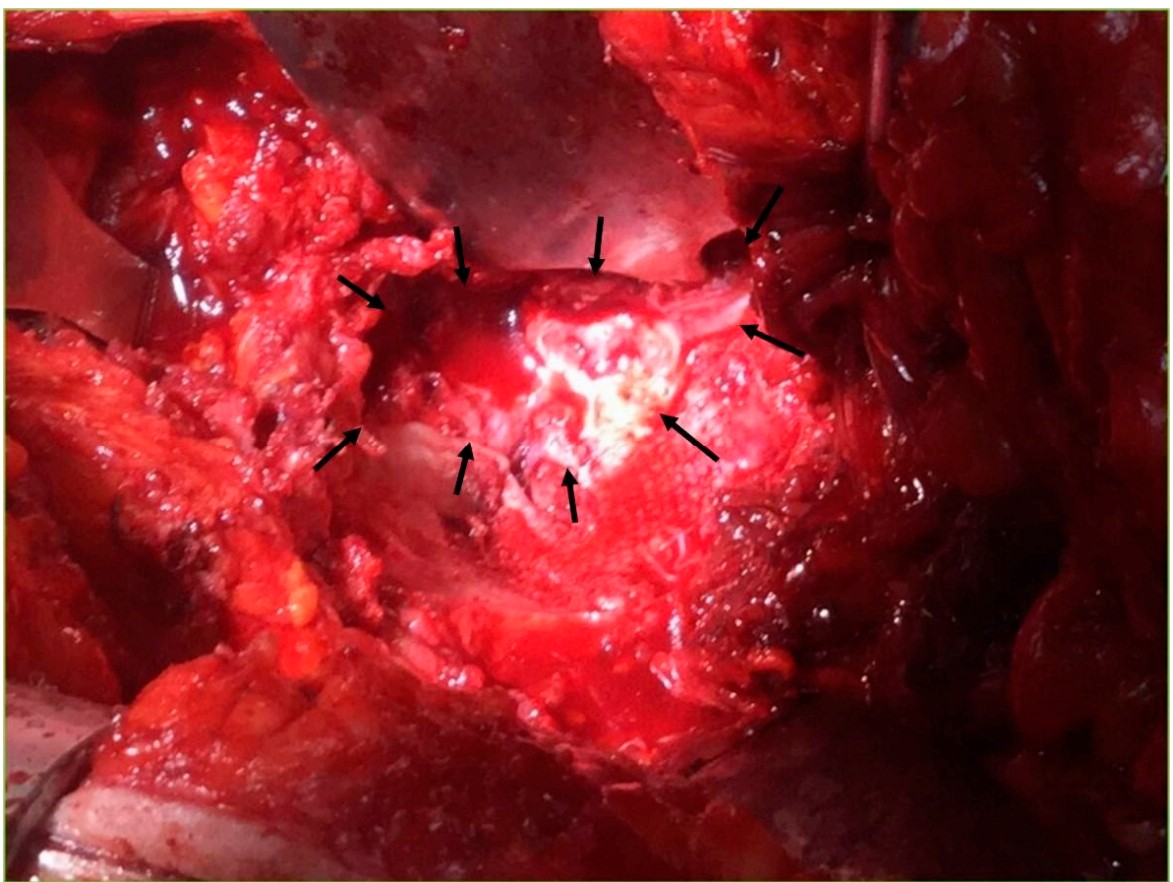

**Figure 3.** Intraoperative view of the massive acetabular bone loss (black arrows).

The microbiological examination of the periprosthetic tissue taken during the surgery led to the diagnosis of PJI by staphylococcus aureus methicillin susceptible (MSSA). The

patient had a history of allergic reactions to drugs, including beta-lactams, sulfonamides, and salicylates. She was, therefore, treated with daptomycin and gentamicin.

The large loss of bone stock led to postoperative complications: three events of spacer dislocation and limb shortening. The instability of the spacer was treated with a pelvic-condylar cast. At the end of the prescribed antibiotic therapy and the normalisation of the inflammation indices, the patient was a candidate for prosthetic re-implantation. Unfortunately, however, the bone loss of the acetabulum was such that it did not allow reconstruction with traditional revision implants, such as revision acetabular components, "Jumbo" cups, stemmed or cemented acetabular components, or reconstruction ring with or without impaction grafting. Consequently, a computer tomography (CT) scan of the pelvis was then performed to obtain the exact estimate of the bone loss. The CT scan was then acquired by the engineers and a preliminary proposal of the implant was made, taking into account the leg length discrepancy, the need to restore the centre of rotation of the hip, and the number, position, and length of the screws for its fixation (Figure 4). This was discussed with the surgeon and modified according to the clinical necessity, and then, with the 3D printer, trials were carried out in sterile plastic material which exactly reproduced the definitive porous titanium alloy implant (Figure 5). Exact indications were also given on how to perform the preliminary reaming.

In the "Second Stage", after the removal of the antibiotic spacer, a deep tissue debridement and abundant antiseptic pulsatile lavage were performed. The acetabular bone loss was such that it was necessary to implant a polypropylene mesh fixed with acrylic glue in order to contain the autologous morselized bone graft.

The implant was then covered with defensive antibacterial coating (DAC, NOVAGENIT® Srl, Trento, Italy), a degradable hyaluronic acid gel loaded with vancomycin, in order to reduce the risk of re-infection. Finally, the 3D custom-made cup (Promade 58 mm with CoCr liner for dual mobility, Lima Corporate, San Daniele del Friuli, Italy) was implanted on the patient. The shape of the cup and the direction of the screws corresponded properly to the loss of bone and to the patient's anatomy (Figure 6).

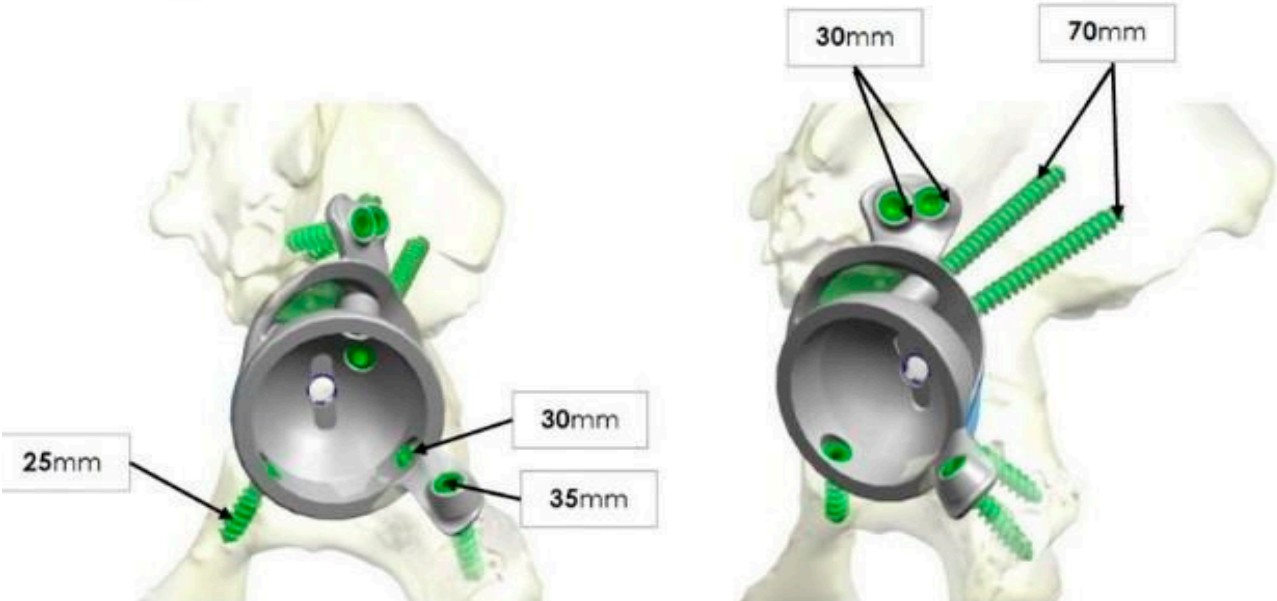

**Figure 4.** Design proposal of the 3D custom-made acetabular cup.

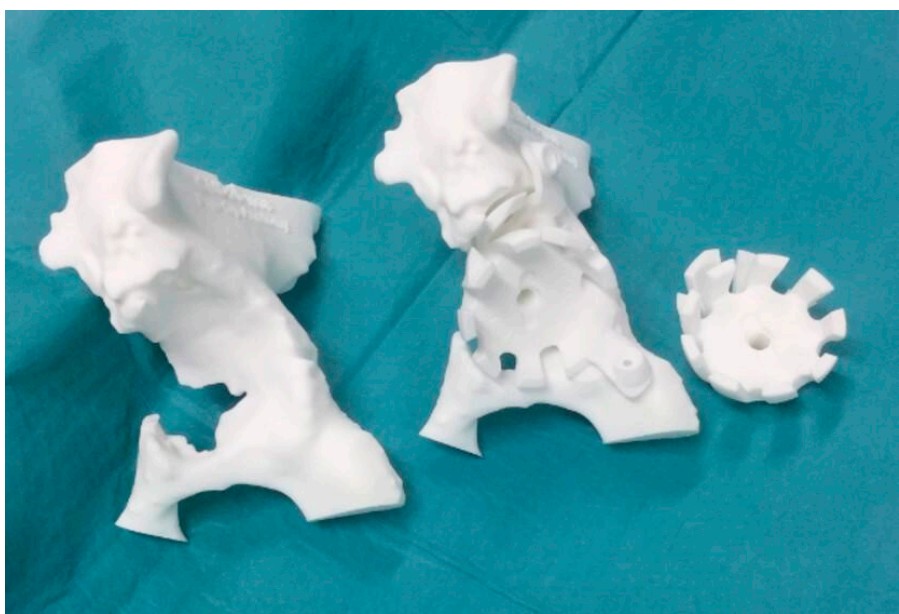

**Figure 5.** Sterile plastic trials used to check the correct positioning of the custom-made cup during surgery.

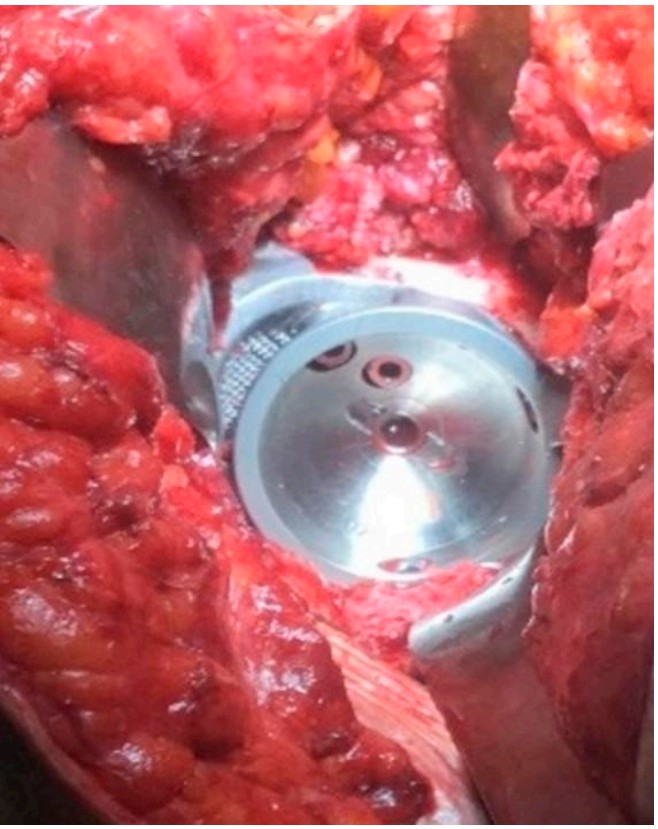

**Figure 6.** Intraoperative view of the implanted custom-made cup that corresponded optimally to the bone loss and to the patient's anatomy.

The previous femoral osteotomy appeared to be well healed, but, as a precaution, four metal cerclages were implanted around the femoral shaft before preparing the canal for the definitive femoral prosthetic implant. A revision femoral stem (modulus hip system size 17 mm/B/125°, Lima Corporate, San Daniele del Friuli, Italy) was implanted together with a dual-mobility coupling (dual mobility liner 42 mm, Lima Corporate, San Daniele

del Friuli, Italy; ball head 28 mm medium, Biolox Delta, Ceramtec, Plochingen, Germany). Antibiotic prophylaxis was continued for 14 days after surgery. The patient assumed the unloaded standing position on the left lower limb on the third postoperative day. The full load was granted 3 months after the surgical procedure, when osseointegration was already visible radiographically. Upon clinical and radiographic evaluation 6 months after the surgical procedure, the implant was well positioned and osseointegrated. The surgical scar was in order, with no signs of infection. The patient did not complain of any painful symptoms and declared a marked improvement in quality of life. At this time, the patient returned to normal activity of daily life without pain or any sign of infection. At a 1 year follow-up, the patient did not complain of any discomfort and also decided to undergo contralateral hip replacement for hip osteoarthritis secondary to DDH (Figure 7).

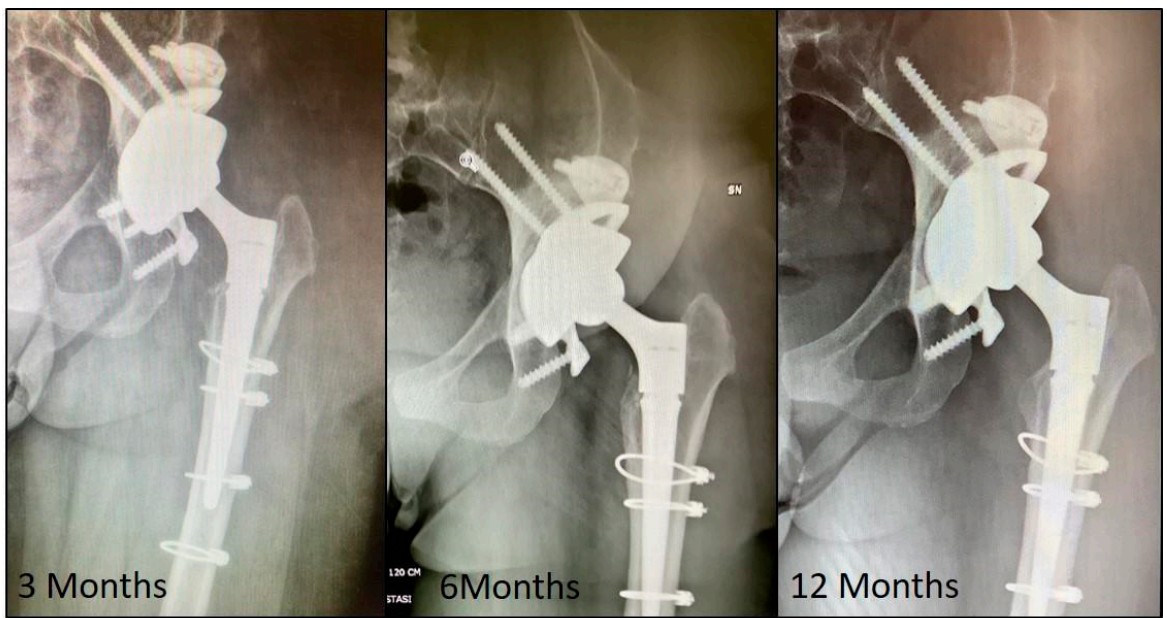

**Figure 7.** X-ray of the hip at 3, 6, and 12 months' follow-up, showing a stable fixation of the implant over the time.

## 4. Discussion

3D-printing technology is innovative and revolutionary in many fields. In orthopaedic surgery, the premises are very interesting, but the application's perspectives are not yet completely well known. Our experience reports some of the possible advantages of this new technology in orthopaedic surgery. Our results in trauma surgery offer a modern proposal to surgeons for the rational planning of open reduction and internal fixation (ORIF) of complex fractures [16]. The lower number of intraoperative radiographic controls, the reduction of the procedure time, and the advanced patient compliance allow the reduction of X-ray exposure for patients and healthcare professionals and optimise surgical outcomes. In addition, the introduction of this method during residency in the orthopaedic surgical area could practically involve the trainees and improve training satisfaction and quality of learning [17].

The development and feasibility of this method, however, depends on the cost and effectiveness analysis. A 3D printing model adds to the already-required costs of a CT, X-ray, or MRI exam. The dimensions of the models, the material used for printing, and the details of the prototypes directly influence the final cost. To date, our experience shows that the major charge for a custom-made prosthesis paid by the public health service can offer a better chance for a new life to a patient who will be able to return autonomously, without burdening additional state subsidies for disability or an inability to work. At the same time, the 15% reduction in surgical time for proximal humeral fractures leads us to conclude that the savings obtained on less use of the active operating room was about 300 euros per

operation [18,19], but we are still not able to provide a suitable follow-up of these patients to certify even better clinical outcomes.

One of the limitations of this method, however, is that the procedures for printing the 3D model, in our experience, have proved to be long and complex. The accuracy of 3D printing depends primarily on the quality of the CT scan, which, for fractures, should be able to create cuts within 0.5 mm, and also on the technical capabilities of the printer. We did not encounter problems in print accuracy due to limited resolution and/or bone segmentation in each slice of the CT. For practical reasons (to optimise printing time and not to print exceeding and quite useless parts of the humerus bone diaphysis) the print layers were on planes spatially rotated with respect to the CT images planes. The "3D model with oriented surfaces creation" during phase three creates a continuous and interpolated surface between the CT sections. As a direct consequence, there is no loss of resolution due to the spatial rotation of the bone's 3D model and the following definition of the printing layers. The loss of parallelism between CT planes and printing planes does not affect the accuracy level of the final 3D-printed object.

Contemporary 3D printers have an extruded filament thickness of approximately 0.1 mm. In practice, due to the vibrations originating from the movement of the printer itself, the spatial resolution that can be obtained rarely reaches this limit, and is usually about 0.5 mm. Another purely surgical limitation is that the 3D-printed fracture simulation was performed without soft tissues so that the implants could be oriented in the most optimal position, but, certainly, this is not totally reproducible in reality.

On the other hand, the management of pelvis bone loss in revision hip prosthetic surgery is often a real challenge for the orthopaedic surgeon [20]. In the past, reconstruction often seemed impossible, with a strong negative impact on the patient's quality of life and with a high negative economic and social impact. Nowadays, the 3D printing technology application to prosthetic surgery has made it possible to overcome these limits. In addition, the in-depth study of the anatomy and the printing of models exactly identical to those required also facilitates the surgical implantation technique. However, it is always necessary to ensure that the porous coating of these implants comes into contact with autologous or homologous bone in order to favour the implant osseointegration. This is, in fact, essential for a good outcome and for the prevention of loosening [21]. In addition, defensive antibacterial coating (DAC) is important to reduce the risk of re-infection [22]. Moreover, 3D custom-made implants permit us to spare the use of bone grafts, lowering costs and the risk of infective disease transmission related to the use of human donors' bone graft transplantation (HIV, HBV, or HCV), assuring at the same time a safer primary stability and fixation of the cup. This may permit a faster recovery and return to normal weight bearing with better clinical outcomes and quality of life, even in the most severe bone loss cases [21].

## 5. Conclusions

The 3D printing technology showed many advantages in orthopaedic surgery for planning and training, but also for addressing large bone stock deficits. The two examples of employment of this study showed the versatility and the effectiveness of 3D printing technologies used in many fields of orthopaedic surgery. Many other usages are possible, and in the future, with more improvements, it will be adopted in normal clinical practice. This new technology allows for a very accurate preoperative plan in traumatology, prosthetic, and oncologic reconstructive surgery. In more complex cases, 3D printing in orthopaedics makes it possible to facilitate and optimise surgery operative time and improve clinical results, with undoubted economic savings and patient satisfaction. Moreover, this technology is helpful for training residents and young surgeons.

3D technology represents an important resource for the future. Medical research is increasingly moving towards the personalization of therapy. In this perspective, the possibility of using devices created based on the anatomy of the individual patient is certainly at the forefront. Nowadays, all this is not used in a systematic way, but it

represents the strategy for the treatment of the most complex cases. Therefore, large-scale systematic studies are needed in order to effectively evaluate the costs, benefits, and risks of this modern and promising resource.

**Author Contributions:** M.S. and A.F. collected data and wrote the manuscript; T.P. developed the engineering design of 3D printing; G.L. critically revised the manuscript. All authors have read and agreed to the published version of the manuscript.

**Funding:** This research received no external funding.

**Institutional Review Board Statement:** This study was conducted in accordance with the Declaration of Helsinki, and approved by the Institutional Review Board of the University of L'Aquila with n°54/2021-2022.

**Informed Consent Statement:** The patients enrolled expressed written informed consent.

**Data Availability Statement:** Not applicable.

**Acknowledgments:** The authors are very grateful to the ViviamoLAQ non-profit association, born after the earthquake that destroyed the city of L'Aquila in 2009, whose members donated 3D-printed PPE to Italian healthcare professionals during the COVID-19 pandemic and collaborated in carrying out this study.

**Conflicts of Interest:** The authors declare no conflict of interest.

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
