# Peer review of "3D Printing Applications in Orthopaedic Surgery: Clinical Experience and Opportunities"

_applsci, doi:10.3390/app12073245_

Round 1

Reviewer 1 Report

This manuscript presents a very interesting case study regarding 3D printed production of fractured bones. The topic is very up-to-date and scientifically of interest. Moreover, it addresses a very urgent problem, which is a more effective and successive surgical processes. It is a great contribution to the scientific literature.

My comment is that the Introduction part should be supplemented with more references on the current trend of this research filed and more data on the materials, biopolymers that are commonly used in 3D printing of fractured body parts and their advantages and limitations.

Author Response

Dear reviewer,

we thank you for your suggests and we appreciate very much your remarks that surely will improve the quality of the study.

  • English language and style are fine/minor spell check required

We revised the language

  • The Introduction part should be supplemented with more references on the current trend of this research filed and more data on the materials, biopolymers that are commonly used in 3D printing of fractured body parts and their advantages and limitations.

The introduction is supplemented, more details about materials are reported with relative references:

“3D-printed models seem to be the best way to get agreement whatever the experience of the observers [5]. Typically, these models are built layer by layer with the print hothead deposing material on the segmented area by working with a special plastic resin, such as acrylonitrile butadiene styrene (ABS) material, or polylactic acid (PLA) [5,6]. While ABS is printed with extruder temperatures of around 240 °C and requires a heated print bed, PLA is more manageable, is fully recyclable as it is a bioplastic derived from corn or sugar cane, and does not emit toxic fumes during printing.

The 3D plastic models aim is to improve the diagnosis of fractures and to help the surgeon think ahead about the surgical approach and hardware needed. Three-dimensional printing is currently under development around the world for orthopaedic disorders, congenital paediatric orthopaedic problems or tumours [7-9].”

Dear reviewer, we thank you for your suggests and we appreciate very much your remarks that surely will improve the quality of the study.

  • English language and style are fine/minor spell check required

We revised the language

  • The Introduction part should be supplemented with more references on the current trend of this research filed and more data on the materials, biopolymers that are commonly used in 3D printing of fractured body parts and their advantages and limitations.

The introduction is supplemented, more details about materials are reported with relative references:

“3D-printed models seem to be the best way to get agreement whatever the experience of the observers [5]. Typically, these models are built layer by layer with the print hothead deposing material on the segmented area by working with a special plastic resin, such as acrylonitrile butadiene styrene (ABS) material, or polylactic acid (PLA) [5,6]. While ABS is printed with extruder temperatures of around 240 °C and requires a heated print bed, PLA is more manageable, is fully recyclable as it is a bioplastic derived from corn or sugar cane, and does not emit toxic fumes during printing.

The 3D plastic models aim is to improve the diagnosis of fractures and to help the surgeon think ahead about the surgical approach and hardware needed. Three-dimensional printing is currently under development around the world for orthopaedic disorders, congenital paediatric orthopaedic problems or tumours [7-9].”

Dear reviewer, we thank you for your suggests and we appreciate very much your remarks that surely will improve the quality of the study.

  • English language and style are fine/minor spell check required

We revised the language

  • The Introduction part should be supplemented with more references on the current trend of this research filed and more data on the materials, biopolymers that are commonly used in 3D printing of fractured body parts and their advantages and limitations.

The introduction is supplemented, more details about materials are reported with relative references:

“3D-printed models seem to be the best way to get agreement whatever the experience of the observers [5]. Typically, these models are built layer by layer with the print hothead deposing material on the segmented area by working with a special plastic resin, such as acrylonitrile butadiene styrene (ABS) material, or polylactic acid (PLA) [5,6]. While ABS is printed with extruder temperatures of around 240 °C and requires a heated print bed, PLA is more manageable, is fully recyclable as it is a bioplastic derived from corn or sugar cane, and does not emit toxic fumes during printing.

The 3D plastic models aim is to improve the diagnosis of fractures and to help the surgeon think ahead about the surgical approach and hardware needed. Three-dimensional printing is currently under development around the world for orthopaedic disorders, congenital paediatric orthopaedic problems or tumours [7-9].”

Dear reviewer, we thank you for your suggests and we appreciate very much your remarks that surely will improve the quality of the study.

  • English language and style are fine/minor spell check required

We revised the language

  • The Introduction part should be supplemented with more references on the current trend of this research filed and more data on the materials, biopolymers that are commonly used in 3D printing of fractured body parts and their advantages and limitations.

The introduction is supplemented, more details about materials are reported with relative references:

“3D-printed models seem to be the best way to get agreement whatever the experience of the observers [5]. Typically, these models are built layer by layer with the print hothead deposing material on the segmented area by working with a special plastic resin, such as acrylonitrile butadiene styrene (ABS) material, or polylactic acid (PLA) [5,6]. While ABS is printed with extruder temperatures of around 240 °C and requires a heated print bed, PLA is more manageable, is fully recyclable as it is a bioplastic derived from corn or sugar cane, and does not emit toxic fumes during printing.

The 3D plastic models aim is to improve the diagnosis of fractures and to help the surgeon think ahead about the surgical approach and hardware needed. Three-dimensional printing is currently under development around the world for orthopaedic disorders, congenital paediatric orthopaedic problems or tumours [7-9].”

Dear reviewer, we thank you for your suggests and we appreciate very much your remarks that surely will improve the quality of the study.

  • English language and style are fine/minor spell check required

We revised the language

  • The Introduction part should be supplemented with more references on the current trend of this research filed and more data on the materials, biopolymers that are commonly used in 3D printing of fractured body parts and their advantages and limitations.

The introduction is supplemented, more details about materials are reported with relative references:

“3D-printed models seem to be the best way to get agreement whatever the experience of the observers [5]. Typically, these models are built layer by layer with the print hothead deposing material on the segmented area by working with a special plastic resin, such as acrylonitrile butadiene styrene (ABS) material, or polylactic acid (PLA) [5,6]. While ABS is printed with extruder temperatures of around 240 °C and requires a heated print bed, PLA is more manageable, is fully recyclable as it is a bioplastic derived from corn or sugar cane, and does not emit toxic fumes during printing.

The 3D plastic models aim is to improve the diagnosis of fractures and to help the surgeon think ahead about the surgical approach and hardware needed. Three-dimensional printing is currently under development around the world for orthopaedic disorders, congenital paediatric orthopaedic problems or tumours [7-9].”

Dear reviewer, we thank you for your suggests and we appreciate very much your remarks that surely will improve the quality of the study.

  • English language and style are fine/minor spell check required

We revised the language

  • The Introduction part should be supplemented with more references on the current trend of this research filed and more data on the materials, biopolymers that are commonly used in 3D printing of fractured body parts and their advantages and limitations.

The introduction is supplemented, more details about materials are reported with relative references:

“3D-printed models seem to be the best way to get agreement whatever the experience of the observers [5]. Typically, these models are built layer by layer with the print hothead deposing material on the segmented area by working with a special plastic resin, such as acrylonitrile butadiene styrene (ABS) material, or polylactic acid (PLA) [5,6]. While ABS is printed with extruder temperatures of around 240 °C and requires a heated print bed, PLA is more manageable, is fully recyclable as it is a bioplastic derived from corn or sugar cane, and does not emit toxic fumes during printing.

The 3D plastic models aim is to improve the diagnosis of fractures and to help the surgeon think ahead about the surgical approach and hardware needed. Three-dimensional printing is currently under development around the world for orthopaedic disorders, congenital paediatric orthopaedic problems or tumours [7-9].”

Dear reviewer, we thank you for your suggests and we appreciate very much your remarks that surely will improve the quality of the study.

  • English language and style are fine/minor spell check required

We revised the language

  • The Introduction part should be supplemented with more references on the current trend of this research filed and more data on the materials, biopolymers that are commonly used in 3D printing of fractured body parts and their advantages and limitations.

The introduction is supplemented, more details about materials are reported with relative references:

“3D-printed models seem to be the best way to get agreement whatever the experience of the observers [5]. Typically, these models are built layer by layer with the print hothead deposing material on the segmented area by working with a special plastic resin, such as acrylonitrile butadiene styrene (ABS) material, or polylactic acid (PLA) [5,6]. While ABS is printed with extruder temperatures of around 240 °C and requires a heated print bed, PLA is more manageable, is fully recyclable as it is a bioplastic derived from corn or sugar cane, and does not emit toxic fumes during printing.

The 3D plastic models aim is to improve the diagnosis of fractures and to help the surgeon think ahead about the surgical approach and hardware needed. Three-dimensional printing is currently under development around the world for orthopaedic disorders, congenital paediatric orthopaedic problems or tumours [7-9].”

Dear reviewer, we thank you for your suggests and we appreciate very much your remarks that surely will improve the quality of the study.

  • English language and style are fine/minor spell check required

We revised the language

  • The Introduction part should be supplemented with more references on the current trend of this research filed and more data on the materials, biopolymers that are commonly used in 3D printing of fractured body parts and their advantages and limitations.

The introduction is supplemented, more details about materials are reported with relative references:

“3D-printed models seem to be the best way to get agreement whatever the experience of the observers [5]. Typically, these models are built layer by layer with the print hothead deposing material on the segmented area by working with a special plastic resin, such as acrylonitrile butadiene styrene (ABS) material, or polylactic acid (PLA) [5,6]. While ABS is printed with extruder temperatures of around 240 °C and requires a heated print bed, PLA is more manageable, is fully recyclable as it is a bioplastic derived from corn or sugar cane, and does not emit toxic fumes during printing.

The 3D plastic models aim is to improve the diagnosis of fractures and to help the surgeon think ahead about the surgical approach and hardware needed. Three-dimensional printing is currently under development around the world for orthopaedic disorders, congenital paediatric orthopaedic problems or tumours [7-9].”

Dear reviewer, we thank you for your suggests and we appreciate very much your remarks that surely will improve the quality of the study.

  • English language and style are fine/minor spell check required

We revised the language

  • The Introduction part should be supplemented with more references on the current trend of this research filed and more data on the materials, biopolymers that are commonly used in 3D printing of fractured body parts and their advantages and limitations.

The introduction is supplemented, more details about materials are reported with relative references:

“3D-printed models seem to be the best way to get agreement whatever the experience of the observers [5]. Typically, these models are built layer by layer with the print hothead deposing material on the segmented area by working with a special plastic resin, such as acrylonitrile butadiene styrene (ABS) material, or polylactic acid (PLA) [5,6]. While ABS is printed with extruder temperatures of around 240 °C and requires a heated print bed, PLA is more manageable, is fully recyclable as it is a bioplastic derived from corn or sugar cane, and does not emit toxic fumes during printing.

The 3D plastic models aim is to improve the diagnosis of fractures and to help the surgeon think ahead about the surgical approach and hardware needed. Three-dimensional printing is currently under development around the world for orthopaedic disorders, congenital paediatric orthopaedic problems or tumours [7-9].”

Dear reviewer, we thank you for your suggests and we appreciate very much your remarks that surely will improve the quality of the study.

  • English language and style are fine/minor spell check required

We revised the language

  • The Introduction part should be supplemented with more references on the current trend of this research filed and more data on the materials, biopolymers that are commonly used in 3D printing of fractured body parts and their advantages and limitations.

The introduction is supplemented, more details about materials are reported with relative references:

“3D-printed models seem to be the best way to get agreement whatever the experience of the observers [5]. Typically, these models are built layer by layer with the print hothead deposing material on the segmented area by working with a special plastic resin, such as acrylonitrile butadiene styrene (ABS) material, or polylactic acid (PLA) [5,6]. While ABS is printed with extruder temperatures of around 240 °C and requires a heated print bed, PLA is more manageable, is fully recyclable as it is a bioplastic derived from corn or sugar cane, and does not emit toxic fumes during printing.

The 3D plastic models aim is to improve the diagnosis of fractures and to help the surgeon think ahead about the surgical approach and hardware needed. Three-dimensional printing is currently under development around the world for orthopaedic disorders, congenital paediatric orthopaedic problems or tumours [7-9].”

Dear reviewer, we thank you for your suggests and we appreciate very much your remarks that surely will improve the quality of the study.

  • English language and style are fine/minor spell check required

We revised the language

  • The Introduction part should be supplemented with more references on the current trend of this research filed and more data on the materials, biopolymers that are commonly used in 3D printing of fractured body parts and their advantages and limitations.

The introduction is supplemented, more details about materials are reported with relative references:

“3D-printed models seem to be the best way to get agreement whatever the experience of the observers [5]. Typically, these models are built layer by layer with the print hothead deposing material on the segmented area by working with a special plastic resin, such as acrylonitrile butadiene styrene (ABS) material, or polylactic acid (PLA) [5,6]. While ABS is printed with extruder temperatures of around 240 °C and requires a heated print bed, PLA is more manageable, is fully recyclable as it is a bioplastic derived from corn or sugar cane, and does not emit toxic fumes during printing.

The 3D plastic models aim is to improve the diagnosis of fractures and to help the surgeon think ahead about the surgical approach and hardware needed. Three-dimensional printing is currently under development around the world for orthopaedic disorders, congenital paediatric orthopaedic problems or tumours [7-9].”

Dear reviewer, we thank you for your suggests and we appreciate very much your remarks that surely will improve the quality of the study.

  • English language and style are fine/minor spell check required

We revised the language

  • The Introduction part should be supplemented with more references on the current trend of this research filed and more data on the materials, biopolymers that are commonly used in 3D printing of fractured body parts and their advantages and limitations.

The introduction is supplemented, more details about materials are reported with relative references:

“3D-printed models seem to be the best way to get agreement whatever the experience of the observers [5]. Typically, these models are built layer by layer with the print hothead deposing material on the segmented area by working with a special plastic resin, such as acrylonitrile butadiene styrene (ABS) material, or polylactic acid (PLA) [5,6]. While ABS is printed with extruder temperatures of around 240 °C and requires a heated print bed, PLA is more manageable, is fully recyclable as it is a bioplastic derived from corn or sugar cane, and does not emit toxic fumes during printing.

The 3D plastic models aim is to improve the diagnosis of fractures and to help the surgeon think ahead about the surgical approach and hardware needed. Three-dimensional printing is currently under development around the world for orthopaedic disorders, congenital paediatric orthopaedic problems or tumours [7-9].”

Dear reviewer, we thank you for your suggests and we appreciate very much your remarks that surely will improve the quality of the study.

  • English language and style are fine/minor spell check required

We revised the language

  • The Introduction part should be supplemented with more references on the current trend of this research filed and more data on the materials, biopolymers that are commonly used in 3D printing of fractured body parts and their advantages and limitations.

The introduction is supplemented, more details about materials are reported with relative references:

“3D-printed models seem to be the best way to get agreement whatever the experience of the observers [5]. Typically, these models are built layer by layer with the print hothead deposing material on the segmented area by working with a special plastic resin, such as acrylonitrile butadiene styrene (ABS) material, or polylactic acid (PLA) [5,6]. While ABS is printed with extruder temperatures of around 240 °C and requires a heated print bed, PLA is more manageable, is fully recyclable as it is a bioplastic derived from corn or sugar cane, and does not emit toxic fumes during printing.

The 3D plastic models aim is to improve the diagnosis of fractures and to help the surgeon think ahead about the surgical approach and hardware needed. Three-dimensional printing is currently under development around the world for orthopaedic disorders, congenital paediatric orthopaedic problems or tumours [7-9].”

Dear reviewer, we thank you for your suggests and we appreciate very much your remarks that surely will improve the quality of the study.

  • English language and style are fine/minor spell check required

We revised the language

  • The Introduction part should be supplemented with more references on the current trend of this research filed and more data on the materials, biopolymers that are commonly used in 3D printing of fractured body parts and their advantages and limitations.

The introduction is supplemented, more details about materials are reported with relative references:

“3D-printed models seem to be the best way to get agreement whatever the experience of the observers [5]. Typically, these models are built layer by layer with the print hothead deposing material on the segmented area by working with a special plastic resin, such as acrylonitrile butadiene styrene (ABS) material, or polylactic acid (PLA) [5,6]. While ABS is printed with extruder temperatures of around 240 °C and requires a heated print bed, PLA is more manageable, is fully recyclable as it is a bioplastic derived from corn or sugar cane, and does not emit toxic fumes during printing.

The 3D plastic models aim is to improve the diagnosis of fractures and to help the surgeon think ahead about the surgical approach and hardware needed. Three-dimensional printing is currently under development around the world for orthopaedic disorders, congenital paediatric orthopaedic problems or tumours [7-9].”

Dear reviewer, we thank you for your suggests and we appreciate very much your remarks that surely will improve the quality of the study.

  • English language and style are fine/minor spell check required

We revised the language

  • The Introduction part should be supplemented with more references on the current trend of this research filed and more data on the materials, biopolymers that are commonly used in 3D printing of fractured body parts and their advantages and limitations.

The introduction is supplemented, more details about materials are reported with relative references:

“3D-printed models seem to be the best way to get agreement whatever the experience of the observers [5]. Typically, these models are built layer by layer with the print hothead deposing material on the segmented area by working with a special plastic resin, such as acrylonitrile butadiene styrene (ABS) material, or polylactic acid (PLA) [5,6]. While ABS is printed with extruder temperatures of around 240 °C and requires a heated print bed, PLA is more manageable, is fully recyclable as it is a bioplastic derived from corn or sugar cane, and does not emit toxic fumes during printing.

The 3D plastic models aim is to improve the diagnosis of fractures and to help the surgeon think ahead about the surgical approach and hardware needed. Three-dimensional printing is currently under development around the world for orthopaedic disorders, congenital paediatric orthopaedic problems or tumours [7-9].”

Dear reviewer, we thank you for your suggests and we appreciate very much your remarks that surely will improve the quality of the study.

  • English language and style are fine/minor spell check required

We revised the language

  • The Introduction part should be supplemented with more references on the current trend of this research filed and more data on the materials, biopolymers that are commonly used in 3D printing of fractured body parts and their advantages and limitations.

The introduction is supplemented, more details about materials are reported with relative references:

“3D-printed models seem to be the best way to get agreement whatever the experience of the observers [5]. Typically, these models are built layer by layer with the print hothead deposing material on the segmented area by working with a special plastic resin, such as acrylonitrile butadiene styrene (ABS) material, or polylactic acid (PLA) [5,6]. While ABS is printed with extruder temperatures of around 240 °C and requires a heated print bed, PLA is more manageable, is fully recyclable as it is a bioplastic derived from corn or sugar cane, and does not emit toxic fumes during printing.

The 3D plastic models aim is to improve the diagnosis of fractures and to help the surgeon think ahead about the surgical approach and hardware needed. Three-dimensional printing is currently under development around the world for orthopaedic disorders, congenital paediatric orthopaedic problems or tumours [7-9].”

Dear reviewer, we thank you for your suggests and we appreciate very much your remarks that surely will improve the quality of the study.

  • English language and style are fine/minor spell check required

We revised the language

  • The Introduction part should be supplemented with more references on the current trend of this research filed and more data on the materials, biopolymers that are commonly used in 3D printing of fractured body parts and their advantages and limitations.

The introduction is supplemented, more details about materials are reported with relative references:

“3D-printed models seem to be the best way to get agreement whatever the experience of the observers [5]. Typically, these models are built layer by layer with the print hothead deposing material on the segmented area by working with a special plastic resin, such as acrylonitrile butadiene styrene (ABS) material, or polylactic acid (PLA) [5,6]. While ABS is printed with extruder temperatures of around 240 °C and requires a heated print bed, PLA is more manageable, is fully recyclable as it is a bioplastic derived from corn or sugar cane, and does not emit toxic fumes during printing.

The 3D plastic models aim is to improve the diagnosis of fractures and to help the surgeon think ahead about the surgical approach and hardware needed. Three-dimensional printing is currently under development around the world for orthopaedic disorders, congenital paediatric orthopaedic problems or tumours [7-9].”

Dear reviewer, we thank you for your suggests and we appreciate very much your remarks that surely will improve the quality of the study.

  • English language and style are fine/minor spell check required

We revised the language

  • The Introduction part should be supplemented with more references on the current trend of this research filed and more data on the materials, biopolymers that are commonly used in 3D printing of fractured body parts and their advantages and limitations.

The introduction is supplemented, more details about materials are reported with relative references:

“3D-printed models seem to be the best way to get agreement whatever the experience of the observers [5]. Typically, these models are built layer by layer with the print hothead deposing material on the segmented area by working with a special plastic resin, such as acrylonitrile butadiene styrene (ABS) material, or polylactic acid (PLA) [5,6]. While ABS is printed with extruder temperatures of around 240 °C and requires a heated print bed, PLA is more manageable, is fully recyclable as it is a bioplastic derived from corn or sugar cane, and does not emit toxic fumes during printing. The 3D plastic models aim is to improve the diagnosis of fractures and to help the surgeon think ahead about the surgical approach and hardware needed. Three-dimensional printing is currently under development around the world for orthopaedic disorders, congenital paediatric orthopaedic problems or tumours [7-9].”

Reviewer 2 Report

Dear Authors, the suggestions have been made directly in the PDF Document. Your article is very well and has been mostly organized well. The topic is important. However only to repeat, the in my view most important things to do. It should be shaped about that your method is of clear advantage, and what is necessary to prove your assumptions by further research, if not proven in this research yet.

Author Response

Dear reviewer,

we thank you for your suggests and we appreciate very much your remarks that surely will improve the quality of the study.

  • Moderate English changes required

We revised the language

  • Multiple formal errors in the text to be corrected

We correct them

  • Abstract-conclusion line 27: what shall be optimized?

“…traumatology in order to optimize surgical procedures and outcomes”

  • Introduction line 37

More detailed information is given

  • Section 1 /Section 2 should be let out

We did it

  • Line 121-122: We added reference
  • Line 292

Human bone grafts can carry infectious diseases, like blood and derivatives.

  • Especially in this part it is a bit unclear that there are real advantages of 3D-printing that otherwise could not be achieved in the classical form. In my opinion this should be explained and argued better to convince the management and society that this leads to probable potential future advantages: The systemic explanation is personalisation, and maybe to find also literature to this topic. So try to explain the difference between the old and new paradigm from the aspect of personalisation. Shape out also long term effects and problems with this from several research perspectives. In the next section (Conclusion --> Conclusion and Outlook) or here and there, shape also what are open research topics for future research in the field.

We added: “The 3D technology represents an important resource for the future. Medical research is increasingly moving towards the personalization of therapy. In this perspective, the possibility of using devices created on the anatomy of the individual patient is certainly at the forefront. Nowadays, all this is not used in a systematic way, but represents the strategy for the treatment of the most complex cases. Therefore, large-scale systematic studies are needed in order to effectively evaluate the costs, benefits and risks of this modern and promising resource.”

This manuscript is a resubmission of an earlier submission. The following is a list of the peer review reports and author responses from that submission.

Round 1

Reviewer 1 Report

Super exciting and vital topic. The quality of language and way of presentation (images) is good. However, I would suggest simply making two papers out of it. 

Aspect 1 is an entirely exciting study. In my opinion, it makes absolutely no sense to combine it with a case report, which doesn't fit the main Conclusion of your abstract. (lower extremity case report after upper extremity study) Additionally, custom made Acetabular Surgery has been a standard procedure for many years. On the other hand, 3D printed ("homemade") surgery planning has been a novelty in recent years.

Bottom line: I suggest you remove the case report from the paper. Therefore let the reader know more about your findings, statistical methods, etc. After this significant revision, you could resubmit.

List of minor errors:

2.2.3: 3!!!! D printing ...
